# Surface Properties and Biological Activities on Bacteria Cells by Biobased Surfactants for Antifouling Applications

**Maria da Gloria C. da Silva** [1,2], **Maria Eduarda P. da Silva** [3], **Anderson O. de Medeiros** [3], **Hugo M. Meira** [2,3] **and Leonie A. Sarubbo** [1,2,3,*]

1 Northeast Biotechnology Network (RENORBIO), Federal Rural University of Pernambuco, Rua Dom Manoel de Medeiros, s/n, Dois Irmãos, Recife 52171-900, Brazil
2 Instituto Avançado de Tecnologia e Inovação (IATI), Rua Potyra, 31—Prado, Recife 50751-310, Brazil
3 Escola Icam Tech, Universidade Católica de Pernambuco (UNICAP), Rua do Príncipe, n. 526, Boa Vista, Recife 50050-900, Brazil
* Correspondence: leonie.sarubbo@unicap.br; Tel.: +55-81-21194084

**Abstract:** Microfouling is the deposition of inorganic and organic material on surfaces and can cause economic losses. This deposition affects the performance of vessels, causes corrosion, clogging of equipment and contaminates the surfaces of medical items and the surface of machinery that handles food; it is controlled by cleaning products that contain synthetic surfactants in their formulations. Biobased products provide a promising basis to produce sustainable chemicals such as surfactants. In the present study, the biobased surfactants glyceryl laurate and hydroxystearic acid were synthesized and evaluated for stability at different pH values, salinity and temperatures. In addition, bioactivity tests against *Pseudomonas aeruginosa* (UCP 0992) and *Bacillus cereus* (UCP 1516) were also performed. Biobased surfactants glyceryl laurate and hydroxystearic acid showed excellent stability against temperature, pH, salinity and emulsifying activities for different kinds of oils; prevented bacterial adhesion by almost 100%; and affected the production of EPS by both bacteria and their consortium when compared to a synthetic surfactant SDS. The results showed the potential of these substances for application as an alternative antifouling non-biocide.

**Keywords:** natural surfactant; biosurfactant; antimicrobial; antiadhesive; *Pseudomonas aeruginosa*; biofilm; coconut oil

## 1. Introduction

The future shortage of fossil fuels is a major challenge for the maintenance of current technological development, and recent research is adding value to biomass from different sources to produce new types of products in search of sustainability [1,2]. Biobased products, i.e., biomass, are made from renewable materials and provide a promising basis to produce more complex objects, materials and sustainable chemicals such as: biodegradable plastics, lubricants, solvents, surfactants, etc. Within this scenario, the production of surfactants of natural origin has shown high potential for application due to the strong demand for biodegradable products [1].

A surfactant is any molecule that attracts polar and nonpolar substances, reduces the surface tension between them and makes them mix. Because they have two parts of opposite affinity (a hydrophobic tail and a hydrophilic head) they can reduce the interfacial tension between two immiscible liquids or between a liquid and a solid. According to the nature of the hydrophilic molecular polarity, surfactants can be classified as anionic (negatively charged), cationic (positively charged), nonionic (uncharged) and zwitterionic surfactants (cationic and anionic centers attached to the same molecule). These properties allow surfactants to have moisturizing, emulsifying, foaming, floating, suspending, antibacterial, antiadhesive and protective actions. They also aid drug absorption and act as detergents [3–7].

Natural surfactants (biotic) that can be extracted directly from biological sources such as microorganisms, plants and animals are called biosurfactants, while biobased surfactants (BS) are the surfactants derived from these sources. The production of synthetic surfactants (abiotic) is a purely chemical process carried out by the petrochemical industry. The biobased surfactants are synthesized by chemical or enzymatic processes using renewable substrates (biomass) as raw material and can be produced experimentally in the laboratory using suitable precursors and conditions. Unlike synthetic surfactants, the production and use of biobased surfactants can also decrease the greenhouse effect by reducing carbon dioxide. In addition to being more biocompatible and biodegradable than conventional surfactants [8], biobased surfactants can be applied in the areas of food, pharmaceuticals, personal care products, cosmetics, detergents, paints, coatings and other industrial sectors [8–11].

Biosurfactants, often confused with biobased surfactants, differ in that they refer specifically to surfactants that are produced by the biochemical routes of microorganisms and plants; thus, they are proteins, lipids and carbohydrates that have surfactant properties. In these organisms, the molecules are often associated with certain cell regions such as the cell wall or membrane, requiring more separation steps and milder recovery methods than those used to obtain biobased surfactants [12,13].

As for the production of biobased surfactants, the main natural resources used for the synthesis of this type of surfactant are fatty acids, fatty acid methyl esters, vegetable oil triacylglycerols (peanut oil, soybean oil and corn oil), carbohydrates, glycerol and amino acids [8,9]. It is known that surfactants from biotic sources are very efficient when compared to abiotic; however, the cost of extracting natural surfactants is still a bottleneck for the widespread exploitation of this technology [7,14–16].

The search for products that satisfy the consumer market, both in terms of the quality of delivery of the benefit and in the value of sustainability and eco-friendliness, encourages research that seeks surfactants with biotic resources. The current applications of these molecules are diverse, such as: the treatment of water containing per/polyfluoroalkyl (PFAS) substances and heavy metal ions [17]; enhanced oil recovery (EOR) from crude oil [18]; cosmetics production [5,8]; antimicrobial, antibacterial and antibiofilm uses [19–23]; and antifouling [24–26].

Microfouling is deposits of inorganic and organic materials on surfaces, and it can occur naturally or be stimulated by human activity [27]. Microfouling is the first step in the growth of biological incrustations on a consolidated substrate immersed in water. Bacteria form multispecies biofilms with other microorganisms and allow the adhesion of larger organisms which cause macrofouling [28]. In the maritime transport industry, surfaces completely covered with water are surrounded by unwanted organic materials, called microfouling that is nothing more than the accumulation of microorganisms and their metabolites, which form biofilms that can attract other types of more harmful fouling such as mussels and barnacles [29–31]. Biofilms are particularly problematical in desalination plants where they form on high-pressure filtration membranes [28]. In the dairy industry, biofilms can be a source of persistent contamination that leads to food spoilage and can lead to public health concerns, such as outbreaks of foodborne pathogens [32,33]. Problems with biological fouling also include increased fuel consumption, pipe clogging, decreased equipment performance and metal corrosion caused by microorganisms due to hydrogen sulfide formed by sulfate-reducing bacteria [29–31].

Synthetic surfactants are already commonly used to control biofilms by cleaning food-handling surfaces and treating chronic wounds [34]. As for those of natural origin, research shows that there are several molecules that can replace conventional surfactants in these applications [35–40], which would be beneficial because the synthesis of chemical surfactants seriously affects the environment through global warming-related problems that cause climate change, ozone depletion and greenhouse gas emissions [41].

This article explores the use of these biobased surfactants to control the formation and growth of common microfouling on the surface of different materials without harming

the environment. In the composition of antifouling products, their elements are expected to be non-toxic to other non-target organisms. One group of antimicrobial compounds found in nature and considered to have little, or no toxicity is the fatty acids group and its corresponding esters [42]. As these surfactants are present in cosmetic and pharmaceutical formulations, their antimicrobial actions are already known, and this activity is a prerequisite for their use in other areas such as antifouling actives. Although these surfactants can be obtained commercially, they are expensive, making them difficult to access and use in applications that require large quantities. Thus, as they can be synthesized through reused vegetable oils, they can be useful for non-cosmetic, pharmaceutical and food applications.

Thus, the objective of the research was to compare the surfactant properties and the bioactivity in Gram-positive and Gram-negative bacterial cells of two liquid solutions of biobased surfactants glyceryl laurate and hydroxystearic acid produced by the chemical alteration of vegetable oils with an interpretation to the possible use of these surfactants in the control of microfouling in humid environments in different industries.

## 2. Materials and Methods

Hydrogen peroxide, glacial acetic acid and oleic acid were used without further purification. All fine chemicals employed in this study were of the highest purity grade, produced by Sigma-Aldrich (Darmstadt, Germany). Coconut oil was purchased from a local market in the city of Recife, state of Pernambuco, Brazil. Coconut (*Cocos nucifera* L.) oils were used as the sources of fatty acids in the glyceryl laurate reactions. Growth medium: nutrient agar (NA) and nutrient broth (NB) purchased from Kasvi Laboratories (São José dos Pinhais, PR, Brazil). A microplate reader tela 5.7″ (Kasuaki) and its accompanying software were employed as well.

### 2.1. Synthesis of Biobased Surfactants Derived from Vegetable Oil

Glyceryl laurate was synthesized according to the method alkaline-catalyzed trans-esterification reactions [43]. Hydroxystearic acid was produced through the epoxidation reaction, followed by the opening of the epoxide ring in an aqueous acidic medium. In a glass flask equipped with a reflux condenser, a mixture containing 2:1:1 moles of hydrogen peroxide, glacial acetic acid and oleic acid, respectively, was added. The mixture was kept under constant stirring at a temperature of 90 °C for a period of 24 h. Soon after, the mixture was transferred to a beaker and put to rest, allowing the complete separation of the phases and the solidification of the upper fraction. The lower phase was rejected, and the hydroxystearic acid was melted and washed three times with distilled water. At the end of the washes, the hydroxystearic acid was recrystallized with a solution of ethanol and water [44,45].

Surface tension (ST) and critical micelle concentration (CMC) were measured at 25 °C using a K10ST digital tensiometer (Krüss, Hamburg, Germany) equipped with a platinum ring. Biobased surfactant solutions (10 mg·L$^{-1}$) were prepared with distilled water, sterilized in an autoclave at 121 °C for 15 min and used in all subsequent experiments. Sodium dodecyl sulphate (SDS) 1% (*w/v*) was used as the positive control and water was used as the negative control [46].

### 2.2. Emulsification Index

The emulsification index was determined using the method described by Cooper and Goldenberg [47]. Briefly, 2 mL of the biobased surfactant solution was added to a test tube containing equal an amount of a single hydrophobic test compound (petroleum, motor oil, diesel or biodiesel). The tube was vortexed for 2 min at max speed and left to stand for 24 h at room temperature. The emulsification index EI$_{24}$ (Equation (1)) was determined by dividing the height of the emulsified layer (mm) by the total height of the liquid column (mm) and multiplying the result by 100. SDS 1% (*w/v*); sterile distilled water was used as the positive and negative controls, respectively. After 24 h of incubation, the height of the

emulsified layer was measured, and the emulsification index (EI$_{24}$), a measure of emulsion stability, was assessed as follows:

$$\text{(Emulsification index) EI}_{24} = \text{height of emulsion formed/total height} \times 100 \qquad (1)$$

### 2.3. Oil Displacement and Drop Collapse Assay

The oil displacement assay described by Morikawa et al. [48] was used with slight modifications. An oil slick was formed by adding 3 mL of used motor oil to 40 mL of distilled water in a Petri dish (diameter: 90 mm). Next, 500 µL of the biobased surfactant solution was gently dispersed onto the center of the oil slick. After 1 min, the diameter of the clearing zone around the solution of the biobased surfactants was measured. To test whether the surfactant produced could decrease the surface tension between water and hydrophobic surfaces, its ability to collapse a droplet of water was tested as follows: Surfactant solution (20 µL) was pipetted as a droplet onto parafilm "M" laboratory film (American National Can, Chicago, IL, USA), and the flattening and spreading of the droplet on the parafilm surface was followed over 1 min. Subsequently, the diameter of the droplet was recorded. SDS 1% (*w/v*) and sterile distilled water were used as the positive and negative controls, respectively. All the experiments were performed in triplicate.

### 2.4. The pH, Salinity and Temperature Effect

The effect of pH on the ST was evaluated after adjustment of the broth pH to 2, 4, 6, 8, 12 and 13 with 6.0 M NaOH or HCl. The effect of the NaCl concentration was determined by adding salt (0.0 to 20.0%) to the samples. The biobased surfactant solutions were also heated to 100 °C and 121 °C for 20, 40, 60, 100, and 140 min and cooled to room temperature for further analysis. The biobased surfactant solutions were then used for the determination of the ST. These assays were carried out in triplicate and did not vary by more than 5%.

### 2.5. Microorganisms: Growth Conditions

The test was carried out using the strains *Pseudomonas aeruginosa* (UCP 0992) and *Bacillus cereus* (UCP 1516), as well as a consortium formed by them. These species were chosen because they form resistant biofilms in different sectors of the economy, being found in many places from medical devices to industrial pipes in the oil industry. The bacteria were obtained from the Culture Bank of the Centre for Research in Environmental Sciences (NPCIAMB) of the Catholic University of Pernambuco. Bacterial strains were cultured overnight in nutrient broth (NB) at 37 °C for 24 h at 200 rpm and 28 ± 2 °C. Each bacterium and the consortium were then diluted in a phosphate buffer and the optical density (OD) was adjusted to 0.1 density at 600 nm. Biobased surfactant solutions (10 mg·L$^{-1}$) were prepared in a phosphate buffer (PBS), filtered through 0.22 µm membrane filters and stored in glass vials. SDS was used as a positive control and PBS served as a negative control. All tests were done in triplicate.

### 2.6. Antiadhesive Assay

The antiadhesive activity of biobased surfactants was determined according to the method described by Gudiña et al. [49] with minor modifications. The wells of a sterile flat bottom microplate (96-well) were filled with 200 µL of the biobased surfactant solution (10 mg·L$^{-1}$) prepared in a phosphate buffer. The filled wells were incubated for 18 h at 4 °C. After this interval, the wells were washed twice with a PBS buffer. An amount of 200 µL of a bacterial suspension in PBS adjusted to an OD of 0.6 (600 nm) was added to each well and incubated for 24 h at 4 °C. After the incubation period, the medium was aspirated. Non-adherent cells were washed off the wells twice with PBS. The adherent microorganisms were fixed with 200 µL of 99% methanol per well, and after 15 min the plates were emptied and allowed to dry. Then, the plates were stained for 5 min with 200 µL of crystal violet (0.1%) per well, washed twice with PBS, dried and resuspended in 200 µL of acetic acid 33% (*v/v*) for 10 min. The optical density of each well was measured

at 595 nm on a microplate reader. SDS and an NB medium were used as controls during the experiments. These experiments were carried out in triplicate.

Changes in the production of carbohydrates and proteins in the extracellular polymeric substances (EPS) of the bacteria were verified by the enzymatic kit Glucose Liquiform and Total Proteins (LABTEST-Brazil) in that order. Adhered bacteria were recovered and suspended in 50 mL of sterile distilled water (SDW). To separate the extracellular structures from the cells, the bacterial suspensions were manually shaken for 5 min and centrifuged at 10,000× *g* at 4 °C for 20 min to precipitate bacteria from the suspension. The microplate reader was used for the quantification of carbohydrates and proteins.

### 2.7. Statistical Analysis

Normality was tested with Shapiro–Wilk's W test. The assays were analyzed by one-way ANOVA. Post-hoc comparisons were made with Tukey's HSD. All experiments were replicated three times independently. Results were expressed as mean ± standard deviation. Differences in activity were analyzed by ANOVA ($p < 0.05$). The STATISTICA 8 software package (StatSoft) and graphing software OriginPro 9 (OriginLab Corporation, Northampton, MA, USA) were used.

## 3. Results and Discussion

### 3.1. Surface Activities of Biobased Surfactants

The surfactant tension and CMC data for the biobased surfactants corroborate those found by Silva et al. (2019) for the use of these surfactants in macrofouling repellency tests.

Regarding the emulsifying activity (Figure 1A), both biobased surfactants achieved satisfactory results, especially glyceryl laurate, which achieved 100% emulsification for three of the four hydrophobic compounds evaluated. This emulsifying capacity enables the homogeneity and stability of the reaction medium to be maintained without phase formation. This emulsifying capacity is important for the removal of oily scales and other dirt since it demonstrates how the surfactant is capable of forming stable aggregates that can carry molecules of different carbon chains. Regarding the dispersion capacity (Figure 1B), glyceryl laurate achieved 100% dispersion of the oil, whereas hydroxystearic acid only achieved 7%. In any case, hydroxystearic acid showed an emulsifying capacity similar to that of SDS. This is important, since detergent products containing this molecule are used to clean different materials, as well as to remove biofilms, often leading to contamination of water bodies by the release of waste into the environment. The use of vegetable oils for the synthesis of similar materials that can replace those obtained from petroleum derivatives is gaining attention from the chemical and polymer industry. These oils are a renewable platform due to their global availability, low cost, biodegradability and low toxicity to humans [50].

The results of the drop collapse test (Figure 1C) indicated that the glyceryl laurate drop collapsed on the hydrophobic surface like parafilm while the hydroxystearic acid, SDS and water drop remained intact. In this case, glyceryl laurate would be the most suitable for making a surface hydrophilic, as well as for surface cleaning actions. The hydroxystearic acid showed similarity with the SDS, although the latter has a high critical micellar concentration CMC, requiring a higher concentration of surfactant to lower the surface tension of the water. Biological surfactants showed a similar ability to reduce the surface tension of water as polysorbates (Tween 20 and Tween 80), which was in conformance with the results obtained in the drop collapse test. The biosurfactant obtained by corn steep liquor reduced the surface tension of the media by 32 units, while the biosurfactant from *Lactobacillus pentosus* reduced it by about 28 units. In contrast, Tween 20 and Tween 80 were able to reduce the surface tension by 36 and 32 units, respectively [51]. According to some authors, there is a correlation between the ability of the microorganism to produce surfactants and the attainment of high droplet diameters. This effect would be indicative of the ability to reduce the surface tension of the liquid, and low diameters would indicate a lack of ability to reduce the surface tension [52,53].

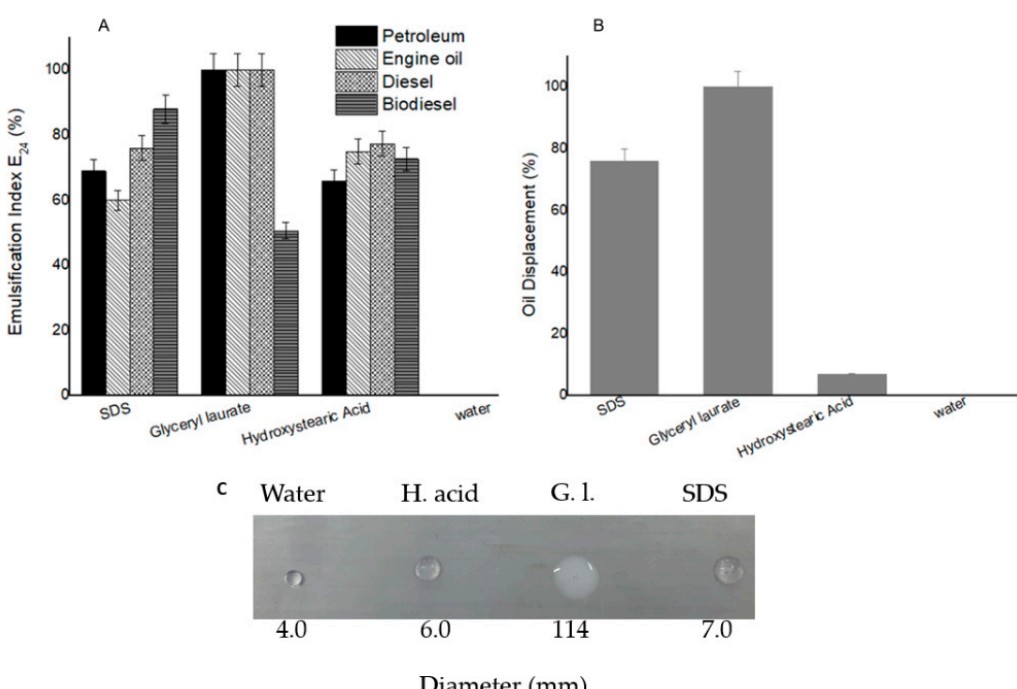

**Figure 1.** (**A**) Values of emulsification index after 24 h, (**B**) engine oil dispersion for biobased surfactants versus conversion surfactant (SDS) and (**C**) drop collapse results. Error bars show standard error from triplicate measurements. Subtitle: H. Acid (hydroxystearic acid); G.L (glyceryl laurate).

### 3.2. Effects of pH, Temperature and Salinity on Surfactant against Tension

Stability studies were performed with the biobased surfactant solutions. Several environmental factors influence the effectiveness of a product. It is therefore important to study these factors when considering the application of biobased surfactants in the environment. The stability of the biobased surfactants with respect to salinity, pH and heating time is shown in Figure 2A–D. The biobased surfactant hydroxystearic acid had a reduction in tension when at pH 6, while glyceryl laurate had an increase in that value from pH 13. As the average salinity of the sea is around 3.5% and the average pH is around 8.0, both biobased surfactants can be used for marine applications. Moreover, both surfactants remained practically stable with extreme changes in pH, salinity and heating time, with glyceryl laurate exhibiting slightly greater stability. Therefore, the tests demonstrate the instantaneous and long-term stability of both surfactants, which increases the shelf life of these natural compounds. Being stable and maintaining properties under different environmental conditions, is a critical requirement to be included in detergent formulations. Research applying the Box–Behnken factorial design to study the effects of pH, temperature and salinity on the surfactant properties of a biosurfactant produced by *Lactobacillus pentosus* observed that the variable that had the greatest effect on the surfactant properties of the biosurfactant was pH. At acidic pH (pH 3–5.5), the decrease in salinity and temperature acted synergistically, reducing the surface tension of the biosurfactant. At pH 8, the same effect on surface tension was observed with increasing salinity and temperature [54]. Many of the environments that suffer fouling have fluctuations in ion concentration, temperature and pH. These changes in physical-chemical parameters may be inherent to the activities performed, and manipulations may be necessary to maintain a low amount of biofouling on the surfaces used. O'Toole et al. (2015) report a systematic study of bacterial biofilm cell death with exposure times ranging from 1 to 30 min in the temperature range of 50 to 80 °C. Related to the development of composite coatings, which are precisely these temperatures on implant surfaces, this work aims to develop a new method to mitigate biofilm infection of medical implants [55].

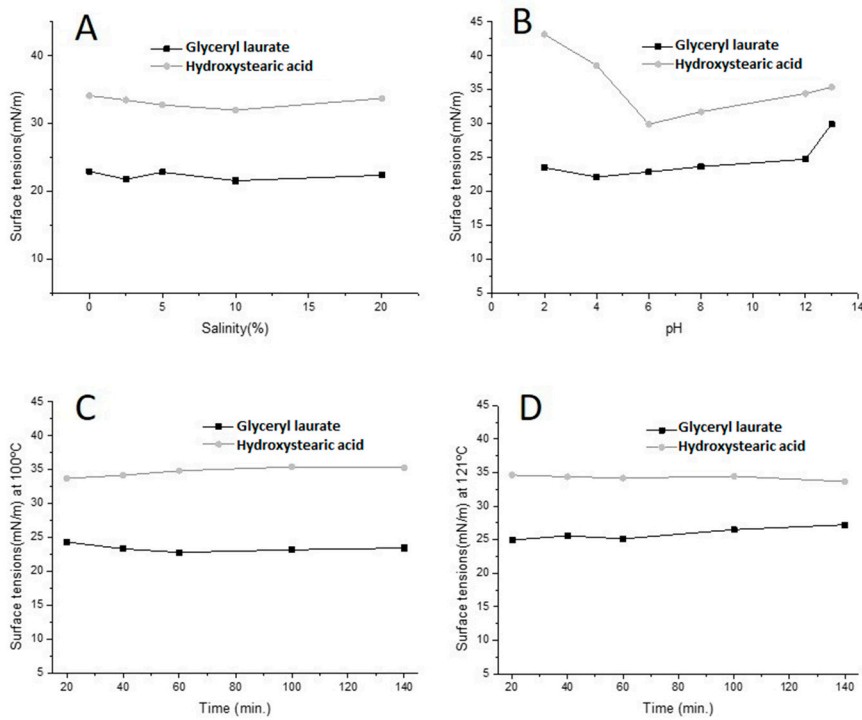

**Figure 2.** Influence of salt concentration (**A**), pH (**B**) and temperature versus time (**C**,**D**) on surface activity of the glyceryl laurate and hydroxylated oleic acid.

### 3.3. Antibacterial Bioactivity and Antiadhesive Assay

The biobased surfactants synthesized exhibited excellent antimicrobial action for both species tested (Figure 3A–C). All surfactants showed some activity regarding the microbial inhibition of the consortium.

The treated groups showed a significant difference in the reduction of adherence and the production of carbohydrates and proteins in comparison to the positive control with culture medium in both strains and in the consortium. The adhesion and production of proteins by bacteria were more affected than carbohydrates. SDS and hydroxystearic acid in terms of total carbohydrate did not show significant differences. SDS and hydroxystearic acid probably affected total carbohydrate production. However, glyceryl laurate presented results that did not differ in relation to the nutrient broth for the Gram-positive strain and the consortium. The same did not occur for the Gram-negative strain when isolated. The treatment with glyceryl laurate was the one that most affected the fixation of the strains. The tested Gram-positive strain was the one that showed more susceptibility to treatments with surfactants. Surfactant molecules, whether natural or synthetic, have the ability to change the surfaces they come in contact with. They generally have the potential to affect both cell–cell and surface–cell interactions [56], and this causes the reduction of interactions and facilitates the detachment of the biofilm when formed. The antimicrobial properties of biosurfactants have been widely reported, along with their ability to prevent adherence and impede the formation of biofilm [57,58]. Biofilm is an envelope of extracellular polymeric substances (EPS). This envelope serves as a strategy for growth, adherence to solid surfaces, survival at adverse conditions, and is a synergy between microbial microconsortia, contributing to the effectiveness of intercellular communication and cooperation and is especially tolerant to biocides and antibiotics [59–62]. EPS contains approximately 1–2% polysaccharide in its matrix and less than this amount in proteins. EPS production increases if bacteria are grown under laboratory conditions with limited nutrients, non-ideal temperature, osmotic stress and other physical factors that restrict growth [63].

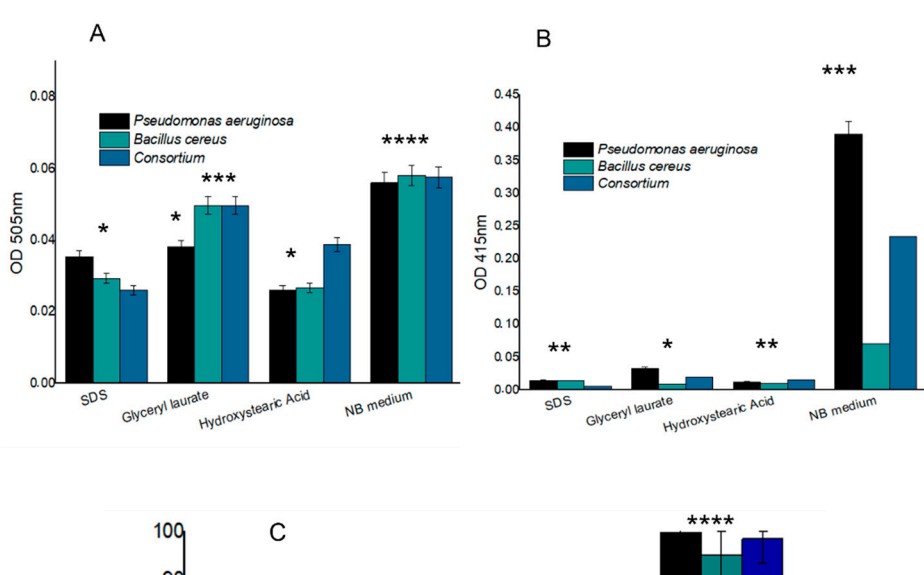

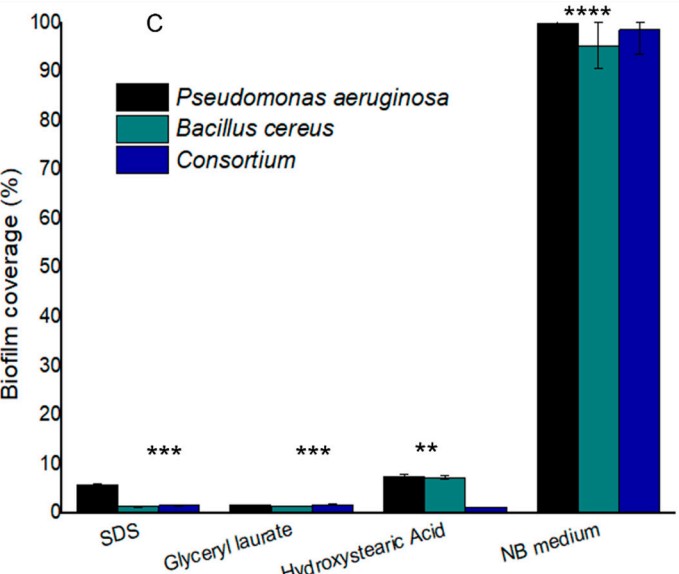

**Figure 3.** Bioactivity assessment of biobased surfactants on Gram-negative and Gram-positive bacteria and their consortium. (**A**) Quantification of the presence of total carbohydrates and (**B**) quantification of the presence of total proteins in a centrifuged sample without a bacterial cell by means of colorimetric enzymatic tests. (**C**) Antiadhesive test with the percentage of well cover with respect to the growth retraction of the tested microorganisms. Error bars represent 95% CI. Significant differences at a = 0.05 in Tukey's test are indicated by a different number of asterisks above the bars.

The exact mechanism by which the biobased surfactants glycerol laurate and hydroxystearic acid are able to kill or inhibit the growth of microorganisms is not known; however, numerous hypotheses have been suggested to explain their general mode of antimicrobial action. They can act as non-ionic surfactants that penetrate and can be incorporated into the bacterial plasma membrane causing structural changes in the cell wall. That results in the release of intracellular constituents: the first being potassium ($K^+$) leakage followed by inorganic phosphates ($P_i$), amino acids and then more significant molecular weight material indicative of gross injury [36,64,65]. Synthetic surfactants are already used for the control of microfouling and macrofouling, as part of the formulation, as well as being used as the main antifouling agent [34]. However, as they are toxic and recalcitrant in very low amounts, they can affect other organisms in the aquatic ecosystem and should be used with caution. In Figure 4A,B, it is shown that the structures of the surfactants are mainly composed of fatty acid monoesters with antimicrobial properties [66]. Studies have shown that metal surfaces preconditioned with biosurfactants are capable of strongly reducing microbial contamination, inhibiting or reducing the development of biofilms [67]

and limiting the occurrence of biofouling on immersed metal structures when incorporated into paints [68]. *P. aeruginosa* PAO1 and *B. cereus*, which are two biofilm-forming bacteria, had their cells and protoplasts disrupted by lipopeptide 6-2 produced by *B. amyloliquefaciens* anti-CA. Lipopeptide 6-2 can also effectively inhibit biofilm formation and disperse pre-formed biofilms. Furthermore, when it was incorporated into a copper-free paint, bacterial adhesion was reduced from 11.4 to 1.8%. According to the authors, Tween 85 is a non-ionic molecule that changes the characteristics of the coated surface. The authors also suggest that electron-donating surfaces, such as varnishes loaded with Tween 85, contribute to reducing bacterial adhesion. The conclusion is that adhesion is lower on low-energy surfaces, which are therefore easier to clean due to the weaker connection at the interface. Thus, the modification of the surface properties to obtain a more hydrophilic surface decreases the adsorption of biofilms, thereby reducing the adhesion of microorganisms [69]. Several other studies highlight the application of surfactants as antifouling agents. Tween 20 enabled the obtaining of a hydrophilic surface, improving the antifouling characteristics of polypropylene micropore membranes for use in bioreactors [70]. The surfactant Arquad 2C-75® was impregnated in hydrogels, which were extremely effective in preventing both microfouling and macrofouling [71]. The product Mexel® 432 is composed of a mixture of surfactants based on aliphatic amines, and its antifouling performance stems from the formation of a thin repellent film that hinders the development of biofouling [72].

**Figure 4.** The structures of the surfactants of glyceryl laurate (**A**) and hydroxystearic acid (**B**).

*P. aeruginosa* and *B. cereus* are among the bacterial species that show high susceptibility to biocide treatments in industries and become resistant biofilms. Thus, in industrial water systems, it is advisable to switch between biocides to keep biofilms within threshold levels. Formulations that can be developed containing biologically based surfactants such as those studied here can help maintain these levels without environmental damage.

## 4. Conclusions

The results of the present study demonstrate that the biobased surfactants glycerol laurate and hydroxystearic acid, in general, have high stability at a wide range of pH levels, high temperatures and salinity. Both surfactants were able to reduce surface tension, the emulsifier oils of different compositions and demonstrated potential applications for the inhibition of micro biofouling by protecting surfaces and reducing microbial adhesion in different industries. This study also indicates the possibility of using biobased surfactants in the composition of antifouling products as an ecological alternative for the control of microfouling on different materials.

**Author Contributions:** Project design, L.A.S. and M.d.G.C.d.S.; methodology, M.d.G.C.d.S., M.E.P.d.S., A.O.d.M. and H.M.M.; validation, L.A.S. and M.d.G.C.d.S.; formal analysis, L.A.S.; investigation, M.d.G.C.d.S., M.E.P.d.S., A.O.d.M. and H.M.M.; resources, L.A.S.; data curation, M.d.G.C.d.S., M.E.P.d.S., A.O.d.M. and H.M.M.; writing—original draft preparation, M.d.G.C.d.S.; writing—review and editing, L.A.S.; visualization, L.A.S.; supervision, L.A.S.; project administration, L.A.S.; funding acquisition, L.A.S. All authors have read and agreed to the published version of the manuscript.

**Funding:** The present study was funded by the following Brazilian fostering agencies: Fundação de Apoio à Ciência e Tecnologia do Estado de Pernambuco (FACEPE), Conselho Nacional de Desenvolvimento Científico e Tecnológico (CNPq), Coordenação de Aperfeiçoamento de Pessoal de Nível Superior (CAPES) (Grant n. Finance Code 001) and Agência Nacional de Energia Elétrica (ANEEL [Electrical Energy Agency]).

**Institutional Review Board Statement:** Not applicable.

**Informed Consent Statement:** Not applicable.

**Data Availability Statement:** Data is contained within the article.

**Acknowledgments:** The authors are grateful to the Laboratories from the Universidade Católica de Pernambuco (UNICAP) and of the Instituto Avançado de Tecnologia e Inovação (IATI), Brazil.

**Conflicts of Interest:** The authors declare no conflict of interest.

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
