# Peer review of "Surface Properties and Biological Activities on Bacteria Cells by Biobased Surfactants for Antifouling Applications"

_surfaces, doi:10.3390/surfaces5030028_

Round 1

Reviewer 1 Report (Previous Reviewer 2)

The authors submitted a previous version of this manuscript to Processes and revised it for submission to Surfaces.  The authors have removed the replicated information from their original manuscript as advised, however, they have not made any attempt to address my concerns about their protein and carbohydrate measurements. 

I feel that there is not enough new information to warrant publication.

Author Response

Dear reviewer, thank you very much for the suggestions. The quality of the manuscript was improved. A brief discussion around the carbohydrates and proteins present in the EPS of biofilms was addressed.

Reviewer 2 Report (Previous Reviewer 1)

The manuscript is well written. Experiments conducted are properly described, The language of the manuscript is good and the discussion adequate. However the orginality and scientific value of the paper is low.

Author Response

Dear reviewer, thank you very much for the suggestions. The quality of the manuscript was improved.

Reviewer 3 Report (New Reviewer)

Authors have evaluated two bio-based surfactants, glyceryl laurate ad hydroxystearic acid, for antifouling applications. I have the following comments to authors:

-  In the introduction section, authors should incorporate these references: “Moldes et al. Biosurfactants: the use of biomolecules in cosmetics and detergents. In New and Future Developments in Microbial Biotechnology and Bioengineering. DOI: https://doi.org/10.1016/B978-0-444-64301-8.00008-1”, as well as “Moldes et al. Synthetic and Bio-Derived Surfactants Versus Microbial Biosurfactants in the Cosmetic Industry: An Overview. Int. J. Mol. Sci. 2021, 22, 2371”, in the lines 62 to 67 (page 2) to reference more this information.

- Why authors test these two bio-based surfactants, glyceryl laurate ad hydroxystearic acid? Indicate advantages and disadvantages with respect to other bio-based surfactants.

- The data are little discussed probably because there are not many studies with bio-based surfactants but the authors should make a comparison for example with biosurfactants, incorporating these works:

Drop collapse test: Rodríguez-López et al. Biological Surfactants vs. Polysorbates: Comparison of Their Emulsifier and Surfactant Properties. Tenside Surf. Det . 55 (2018) 4.

Effect pH, temperature and salinity: Vecino et al. Study of the Synergistic Effects of Salinity, pH, and Temperature on the Surface-Active Properties of Biosurfactants Produced by Lactobacillus pentosus. J. Agric.Food Chem. 2012, 60, 1258−1265.

Antiadhesive assay: Vecino et al. Bioactivity of glycolipopeptide cell-bound biosurfactants against skin pathogens. International Journal of Biological Macromolecules 109 (2018) 971–979.

Author Response

Dear reviewer, thank you very much for the suggestions. The quality of the manuscript was improved. The information and literature options indicated were performed.

Reviewer 4 Report (New Reviewer)

1. The formulas given in figure 4 are incorrect.

2. Information on the mechanism of action of surfactants on bacteria is not provided.

3. Hydroxystearic acid is usually used as the sodium salt. In its acidic form, it changes the acidity of the environment. At the same time, it irritates the skin. Why hydroxystearic acid sodium salt form has not been studied.

Author Response

Dear reviewer, thank you very much for the suggestions. The quality of the manuscript was improved. The images and mechanisms have been addressed as suggested. We did not study the molecule in the way above, precisely because the objective is to test substances that do not present problems that could affect other living beings.

Round 2

Reviewer 1 Report (Previous Reviewer 2)

I have had a look at the manuscript with fresh eyes, but my view that there is insufficient new data to warrant publication has not changed.

The issue I have with the measurement of protein and carbohydrate is that no units of concentration have been given: an absorbance measurement is not concentration, and the differences in the absorbances shown in Fig 3A are small.  However, even if this were corrected my view on the manuscript would not change.

This manuscript is a resubmission of an earlier submission. The following is a list of the peer review reports and author responses from that submission.

Round 1

Reviewer 1 Report

The manuscript Processes-1760053 is focused on the properties of two surfactants such as:  stability against temperature, pH, salinity,  emulsifying and antimicrobial  activity.

The manuscript is well written. Experiments conducted are properly described, The language of the manuscript is good and the discussion adequate.

Here are only little remarks such as: 9,10-dyhidroxy-octadecanoic acid should be 9,10-dihhdroxy-octadecanoic acid, bacteria strains are not always in italics, there is no need to describe the method of sample preparation for NMR assay.

The general opinion is positive and the manuscript is well written. However it is not a scientific work rather educational. Surfactants used are commercially available and used a use as cosmetic ingredient. There is no need to proceed the synthesis and confirmed the structure of compounds, which are known. Therefore the novelty of the manuscript is very little.

Although manuscript is interesting and well written, in my opinion, experiments conducted are basic and not sufficient to be published in Processes. There is not enough research conducted to be published in such Journal as Processes. Authors should improve the scientific value of the manuscript.

Reviewer 2 Report

In addition to the comments below, the data relating to the characterisation of the biobased surfactants (chemical analysis, surface tension and critical micelle concentration) has been reported by the same authors in a recent paper (https://doi.org/10.1016/j.rsma.2021.101854).  Section 2.1 should be changed to refer to the recent paper, section 3.1 (and Figure 1) should be removed and the first paragraph in section 3.2 should be removed as these contain the exact same data as reported previously.  The analytical data presented in the current manuscript are not sufficient and more data were given in the authors' published paper to support the identity of the synthesised compounds.

Other comments

Section 3.4.  I am not convinced of the antibacterial activities of the surfactants as no growth data are presented rather protein and carbohydrate measurements, which might indicate a reduction in these values in the presence of the surfactants.  In figure 4A and B, it is not sufficient that the absorbance values are given, as absorbance is not a unit of concentration.

I recommend showing the structures of the surfactants.